# Design of Machine Learning Solutions to Post-Harvest Classification of Vegetal Species

**Papa Moussa Diop** [1,†]**, Naoki Oshiro** [2,†]**, Morikazu Nakamura** [2,*]**, Jin Takamoto** [3] **and Yuji Nakamura** [3]

1   Graduate School of Engineering and Science, University of the Ryukyus, Okinawa 903-0213, Japan
2   Faculty of Engineering, University of the Ryukyus, Okinawa 903-0213, Japan; n-oshiro@tec.u-ryukyu.ac.jp
3   Media Transport Corporation, Nakagami-gun, Nakagusuku, Okinawa 901-2423, Japan
*   Correspondence: morikazu@ie.u-ryukyu.ac.jp; Tel.: +81-98-895-8715
†   These authors contributed equally to this work.

**Abstract:** This paper presents a machine learning approach to automatically classifying post-harvest vegetal species. Color images of vegetal species were applied to convolutional neural networks (CNNs) and support vector machine (SVM) classifiers. We focused on okra as the target vegetal species and classified it into two quality types. However, our approach could also be applied to other species. The machine learning solution consists of several components, and each design process and its combinations are essential for classification quality. Therefore, we carefully investigated their effects on classification accuracy. Through our experimental evaluation, we confirmed the following: (1) in color space selection, HLG (hue, lightness, and green) and HSL (hue, saturation, and lightness) are essential for vegetal species; (2) suitable preprocessing techniques are required owing to the complexity of the data and noise load; and (3) the diversity extension of learning image data by mixing different datasets obtained under different conditions is quite effective in reducing the overfitting possibility. The results of this study will assist AI practitioners in the design and development of post-harvest classifications based on machine learning.

**Keywords:** vegetal classification; machine learning; support vector machine (SVM); convolutional neural network (CNN); color space

## 1. Introduction

Over the past few decades, machine learning approaches using imagery methodologies have made significant progress in many practical areas, particularly in agriculture, such as machine-learning-based plant species recognition [1,2], plant disease identification [3], and medicinal plant leaf segmentation [4]. The field of agriculture has benefited immensely from the growth of technologies, such as machine learning, the IoT, big data, networking, and computer vision. These technologies have not only paved the way for better farming approaches but also reinforced the backbone of agriculture by introducing more sustainable approaches for harvesting and marketing, and they have contributed to increasing farmers' revenues over time [5,6].

Compared to many other fields, the agricultural domain is subtle and more prone to failure because of several factors that humans cannot control. For example, soil types vary from place to place, and unpredictable weather and rainfall are correlated with the propagation of certain bacteria and pests. Hence, implementing technologies in agriculture is not a simple task, because scientists must ensure the reproducibility of their techniques.

However, using AI-based systems, farmers can save a significant amount of time and achieve economic gains. As proven by Asif et al. regarding the economic impact of introducing climate-smart agriculture (CSA) to farmers [6], rural zones can benefit greatly from developing crop disease detection systems and even post-harvest classification systems.

Several studies have been conducted to demonstrate the effectiveness of machine learning algorithms in detecting and classifying multiple plant diseases [7]. These primarily

use convolutional neural networks (CNNs) [8,9]. The authors of [9] considered the number of CNN layers and attempted to design shallow layers. Outdoor environments can be obstacles to accuracy because of uncertain lighting conditions. Therefore, Ref. [10] focused on plant detection in dynamic outdoor environments. In [11], the authors proposed a method to identify pests by using a residual CNN based on transfer learning because pests are regarded as one of major causes of crop loss worldwide.

Although machine learning researchers have developed many models based on CNNs, there is still room to improve the CNN model design or apply other models in each specific problem domain. In [12], a disease detection technique for corn leaves was developed, in which individual lesion features were extracted by image processing. In [13], a machine learning approach incorporating qualitative feature analysis was developed, which utilized support vector machines (SVMs) and the *k*-nearest neighbor (KNN) algorithm. The study in [14] reviewed remarkable approaches to plant disease detection.

We consider a machine-learning-based post-harvest classification for vegetal species in this paper. In our previous work [15], we proposed a machine learning approach to classify okra into two categories and presented preliminary experimental evaluation results. The machine learning solution consists of several components, and each design process and its combinations are essential for prediction quality. Therefore, we carefully investigate their effects on the classification accuracy in this study.

In this paper, we enhance the method in [15] using color space changes and several other methods to extract useful vegetal features. The experimental evaluation shows the following: (1) in color space selection, HLG and HSL are essential for vegetal species—they are based on combinations of hue, lightness, and green factor for HLG as well as hue, saturation, and lightness for HSL; (2) suitable preprocessing techniques are required owing to the complexity of the data and noise load; and (3) the diversity extension of learning image data by mixing different datasets obtained under different conditions is quite effective in reducing the overfitting possibility. Finally, the images were passed to the CNN and SVM classifiers. Using both classifiers, our model achieved accuracy exceeding 90%. The results of this study will assist AI practitioners in the design and development of post-harvest classifications based on machine learning.

## 2. Post-Harvest Classification

### 2.1. Datasets

This study focused on the okra classification problem. Okra is divided into two quality types, A and B, according to the Japan Agricultural Cooperatives' (JA) standards. Table 1 lists the requirements for these two quality types. Type A refers to okra that meets the highest requirements and is sold at the highest price. Type B refers to low- to mid-range okra sold at lower prices targeting different customers.

**Table 1.** Okra quality types and size classes (relying on JA specification).

| Quality Type | Size Class | Length (cm) | Bent (cm) | Dots | Color |
|---|---|---|---|---|---|
| A | L | 9.5 to 10.5 | ≤0.5 | no dots | dark green |
| | M | 8.0 to 9.5 | | | |
| B | L | 9.5 to 10.5 | ≤1.0 | ≤ten dots | light green |
| | M | 8.0 to 9.5 | | | |

As mentioned previously, the okra image dataset used in this study comprises two quality types: A and B. Their numbers depend on the harvesting results; usually, the number of type A images is greater than that of B. The dataset used in this study was obtained under two conditions. Images of different okra products were captured at different dates and times. The different conditions required to obtain image data affect the classification

process. Therefore, to evaluate this effect, we distinguished the image datasets captured under different conditions using the notations DS1 and DS2.

Table 2 summarizes the data for each quality type used in the experiment. Only minor, but significant, differences were observed between the two subsets. For example, the okra orientation is vertical on DS2 and horizontal on DS1. The lighting conditions also differed between the two subsets. Additionally, okra colorization was different among the two subsets.

**Table 2.** Data sizes in the experiments

| Dataset | Quality Type | Number of Data |
|---------|--------------|----------------|
| DS1 | A | 3293 |
|     | B | 3039 |
| DS2 | A | 252 |
|     | B | 506 |

The images of okra had 640 × 480 pixels, regardless of their object orientation. Furthermore, okra was not necessarily centered within the image frame. All okra images were taken from the same distance and camera angles. Moreover, a white tray was used as a unified background to limit noise and enhance the okra features. Practically, it is difficult to obtain images under the same conditions at different dates and times. To avoid this bias, we generated a new dataset from the raw dataset. In addition, the dataset contains multiple types of background noise. In our experiment, shade constituted a major issue, as it was often mistaken for okra. The shades of the objects in the shooting environment were also included in specific images (Figure 1). These residues in the images are also considered noise because they can potentially interfere with image processing. The data conversion is explained later.

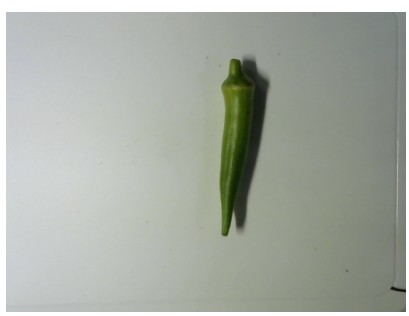

(**a**) Shade as noise    (**b**) Background noises

**Figure 1.** Okra with background noises

## 2.2. Classification Algorithms

Algorithms are critical components of the classification pipeline used in this study. They are described as finite sets of defined sequences that are used to solve problems or classes of problems. These algorithms can be complex and help solve complex problems; however, they always retain a set of logical and unambiguous sequences. Moreover, with the improvements and contributions made in programming languages, such as Python and MATLAB, they are straightforward to implement and design at the user's will. Because our study aims to address classification problems, we focused mainly on the most widely used classifiers for image analysis and classification: ANN and SVM. We narrowed this down to a CNN for ANN. We then compared its performance on accuracy metrics to the SVM classifier and other predominant CNN models (through transfer learning and fine-tuning), such as the residual network (ResNet-50).

### 2.2.1. Convolutional Neural Network

Among the many machine learning algorithms used in image classification, the CNN is one of the most recurrent and remarkably good-performing ones; it implements deep learning to facilitate image processing [9,16,17]. It adds another layer of complexity and hierarchical data representation that was not achievable with traditional machine learning. This progress has been partly made possible by the development of hardware and information processing technology, which allows the utilization of massive amounts of data [9].

In this study, we emphasize preprocessing techniques owing to the complexity of the data and noise included. However, these algorithms are well built and consider several classification factors. Specifically, we compared and evaluated multiple algorithms and tested over 50 possible combinations of CNN architectures to identify the most suitable architecture for our case study. The most important aspects of the CNN architecture that we focused on are as follows:

1.  Diversity within the training data (shuffling, data augmentation, dropout);
2.  Performance metrics (accuracy);
3.  Runtime environments;
4.  Others (learning rate, early stopping).

Small datasets, such as the one used in this study, are susceptible to overfitting. To eliminate this possibility, we combined a set of preprocessing and training techniques. First, we shuffled the data by increasing their size from 2000 to 6332, adding 1000 at every iteration. This shuffling ensured that the dataset remained diverse and unbiased. Moreover, it allowed us to analyze the correlation between the training set size and accuracy. During shuffling, we ensured that the training set was randomly selected from the data pool in every loop. As a second measure to avoid overfitting, a data-augmentation layer was added to the training layers. The principle of data augmentation using the Keras library involves generating new images based on the current training images. Data augmentation can be performed using various parameters. In this study, two parameters were selected: RandomZoom and RandomRotation. Using these parameters, the newly generated images were rotated and zoomed versions of the input images. Therefore, there was greater diversity within the training dataset. Finally, a dropout layer was added to the CNN architecture.

Figure 2 shows the architecture the models used in the experiments. Constructing CNNs in Python offers several possibilities. To enhance the performance of our model, we utilized certain functionalities offered by Keras, such as the learning rate for modulating learning and the early stopping of monitored metrics.

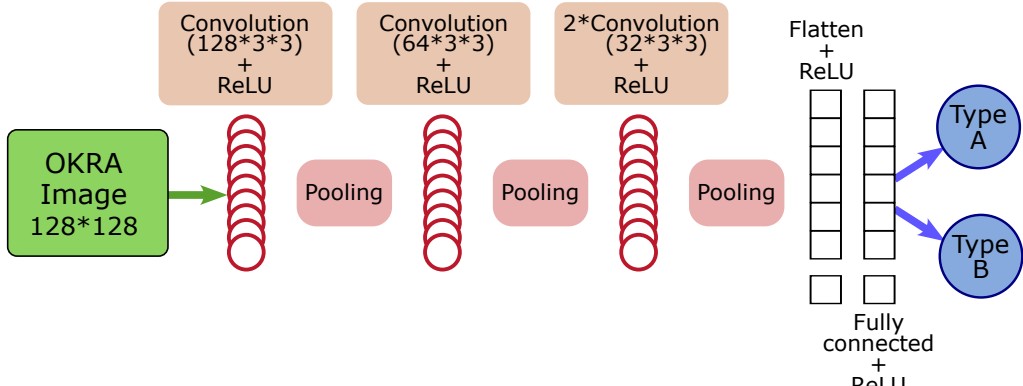

**Figure 2.** CNN model architecture.

### 2.2.2. Support Vector Machine

One classifier that is often compared to neural networks in the leaf recognition literature [18,19] is the SVM. In this study, we compared our designed neural networks to histogram of oriented gradients and support vector machines (HOG-SVMs). HOGs are used for feature reduction; in particular, they are used to lower the problem's complexity

while maintaining as much variation as possible. A HOG, also known as a feature descriptor, is a representation of an image or image patch that simplifies an image by extracting useful information. Therefore, the complexity of the okra input was reduced to 15%, and the classifier only worked with the 3600 most important HOG features. To calculate the HOG, we split the inputs into blocks and grids. Similar to the grids in neural networks, the image can be divided into $6 \times 6$ pixel blocks. The magnitude of the gradient in a given number of directions was calculated for each block. We executed this conversion or transformation by considering an array of data and transforming the items into the final product. This requires grayscale input in advance. Grayscaling is described in the preprocessing section.

We then proceeded to HOGTransformer and finished with StandardScaler. Before proceeding to the HOG transformation, we split our data into test and training sets. We used the train–test–split function from scikit-learn, with 80% of the total set for training and the remaining for the test set. For the SVM, we did not shuffle the data as with the CNN but used the stratified parameter of train–test–split to ensure equal distributions in the training and test sets. The stratified parameter train–test–split ensures that the distribution of classes within the training and testing data is well balanced. Furthermore, the train–test–split function in scikit-learn provides a shuffle parameter to address this issue when performing the split. The random state seeds the shuffle such that it is random but reproducible.

After these preprocessing steps, we used the stochastic gradient descent (SGD) approach to classify okra. SGD is used as a deep learning approach to optimize many convex optimization problems. However, to optimize the model, we compared it with an SVM (SVC) through a grid search using GridSearchCV on scikit-learn. The CV in GridSearchCV stands for cross-validation. This technique combined with a train–test–split allowed us to avoid bias as much as possible. Cross-validation consists of splitting the dataset into *k* folds and each fold is used as a test set, whereas the rest is used as a training set and provides the average. The grid search automates the process of optimizing the parameters. Therefore, rather than manually optimizing the parameters individually, we utilized a grid search and fed it into our parameters to obtain the best parameters for this classification. The parameters used in the grid search were the HOG transformer and classifier. The score method used to evaluate and compare different parameters is the "accuracy" metric. Through this grid search, we obtained the best pipeline parameters within our settings and the highest possible accuracy.

### 2.2.3. Fine-Tuned Convolutional Neural Network

ResNet-50+ is our okra classification version of the ResNet-50 to classify post-harvest okra. ResNet-50 is a convolutional neural network with 50 layers pretrained on the ImageNet database, which contains over a million images of multiple objects. It incorporates many categories of plants and vegetables but not okra. However, the network learned to detect many features within the plants and vegetables, which may be advantageous for this study. The model was fine-tuned to fit the case study. Fine-tuning pretrained models saves programming time and processing resources efficiently because they contain crucial information.

We proceed to fine-tune by "freezing" all model layers except the output layer because it was programmed for tasks specific to the previous model. Therefore, these layers were marked as false. Next, new trainable layers were added before adding an output layer specific to our classification. Our new model adds 10 new layers, including 4 batch normalization layers. After these 10 layers, we added an output layer, which is specific to this classification problem. It is also important to note that our dataset must match the existing model datasets. Particularly, we needed to adapt the size of the images to ResNet's and did not use any other color spaces besides RGB.

*2.3. Image Segmentation*

The goal of segmentation is to simplify and/or change the representation of an image into an input that enhances the features [20]. In this study, we performed segmentation through two processes. The first involved creating a new dataset and the second was used as a preprocessing layer before training the inputs.

2.3.1. Image Segmentation for the Creation of Secondary Datasets

We created a new dataset from raw images using image segmentation techniques. The images were resized, and noise was reduced. We generated a new dataset as follows:

- Grayscaling;
- Blurring;
- Edge detection;
- Center point calculation (from extreme points);
- Largest $x$ and $y$ axis values calculation;
- Attribution of largest values to all inputs.

Grayscaling images evaluate the amount of light on image pixels and convert the RGB image into a gray monochrome image that goes from black for the weakest intensity to white for the most vigorous intensity (this step is further explained in the next section). After this conversion, we blurred the inputs, which removed high-frequency content such as the noise and edges. Blurring the image will allow the machine to detect whatever object is within the frame, rather than external edges.

We applied an edge detection algorithm using the OpenCV Library in Python. The detection function returns a set of four extreme points. These points represent the lowest and highest edges of $x$ and $y$ axes, respectively. Thus, the four edge coordinates of each okra are named right, left, top, and bottom for the highest value on the $x$ axis, the lowest value on the $x$ axis, the highest value on the $y$ axis, and the lowest value on the $y$ axis, respectively. Using $(\text{right} + \text{left})/2$ and $(\text{top} + \text{bottom})/2$, we determined the coordinates of the center point of each okra. We then compared the edge coordinates of all okra and cropped them using the longest distance on the $x$ axis $(\max((\text{right} + \text{left})/2))$ and the longest distance on the $y$ axis $(\max((\text{top} + \text{bottom})/2))$. We have shown two examples of not-centered okras in Figure 3. This segmentation technique centers the okra and simultaneously reduces background noise.

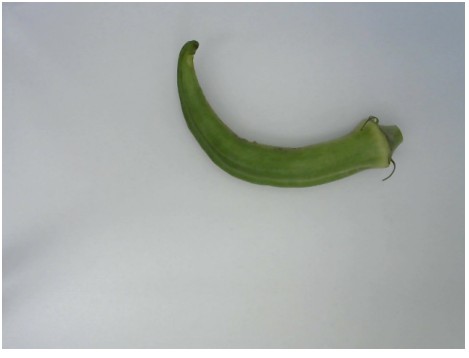 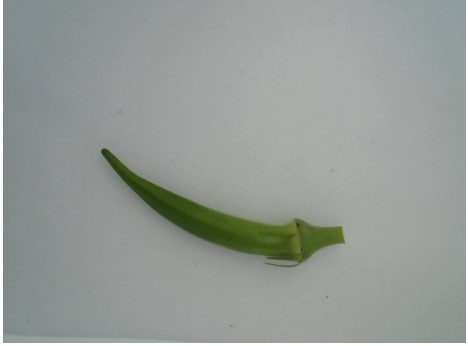

(**a**) Not centered okra       (**b**) Not centered okra

**Figure 3.** Emplacement of okras.

2.3.2. Image Segmentation for the Preprocessing Pipeline

We also applied different image segmentation techniques during preprocessing before training the model. These segmentation techniques reduce background noise and partition images according to their pixel values to detect certain features easily. The $k$-means clustering technique is used for image segmentation. It segments images into several clusters [21]. A single cluster is a set of pixels that are close to each other and different from the other cluster values. In particular, it is a collection of data points aggregated because they share a

particular set of similarities. We start by defining the number $k$, which refers to the number of classes or clusters required. Then, we define the number of iterations of this process and the proximity between points.

This process is repeated 100 times (100 iterations). Moreover, we varied the number of clusters from 2 to 14 with only even numbers, as shown in Figure 4a–c, which include 3 examples. After comparing the number of clusters and their correlation with accuracy, we noticed that beyond six clusters, the accuracy did not improve significantly. It peaked in six clusters in almost all the experiments. Thus, in this study, we define the standard number of clusters as $k = 6$.

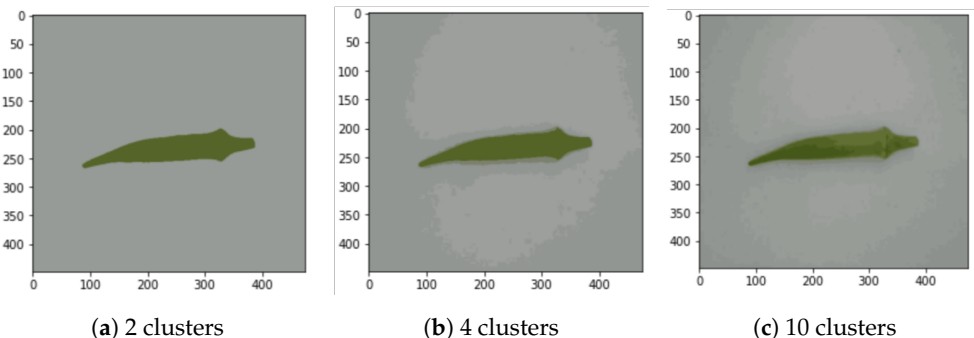

|   (**a**) 2 clusters   |   (**b**) 4 clusters   |   (**c**) 10 clusters   |

**Figure 4.** Segmentation examples with $k$-means for $k = 2$, 4, and 10 (cluster number $k$ was varied from 2 to 14 in the experiment).

### 2.4. Image Preprocessing

Efficient preprocessing helps to detect and emphasize the most critical features. Considering the classification standards set by the Japan Agricultural Cooperatives (JA), we implemented multiple preprocessing techniques and compared them to determine the best fit. The first preprocessing layer consists of image cropping and resizing.

#### 2.4.1. Image Resizing

The purpose of using image resizing is to reduce the number of pixels in an image. Therefore, this process accelerates computation by reducing the number of pixels. However, image resizing may also lead to poor performance. By reducing the number of pixels, we also reduced the number of features or pieces of information contained in the images. Thus, image resizing should be performed carefully, considering time efficiency and performance. In addition, an unknown threshold exists for the image-resizing scale. However, resizing the values proportionally to the initial image preserves the same proportional features. Images used in DS1 and DS2 are $476 \times 465.5$ and $498.5 \times 468$ on average, respectively. For faster computation, the images were resized to $128 \times 128$ pixels. For comparison, we also tested other frames, such as $50 \times 50$ and $64 \times 48$.

#### 2.4.2. Background Noise Removal

The dataset contains multiple types of background noise (Figure 1). In addition, the white tray used as a unifying background contained patterns and textures that could be detrimental to the feature detection of okra. Therefore, noise must be reduced without infringing on proper feature detection. To remove maximum noise, we proceeded as follows:

- Grayscaling;
- Binary transformation;
- Edge detection (of okras).

The grayscale conversion of the images into geometrical data was implemented to optimize the contrast and intensity of the images [4]. The initial color image shown in Figure 5a is converted into a grayscale image, as shown in Figure 5b. The thresholding process creates a binary image from the grayscaled image to translate the value of the image

to its closest threshold. Therefore, it has either one or two possible values for each pixel, as shown in Figure 5c. Grayscaling also reduced the noise compared to the images in Figure 1.

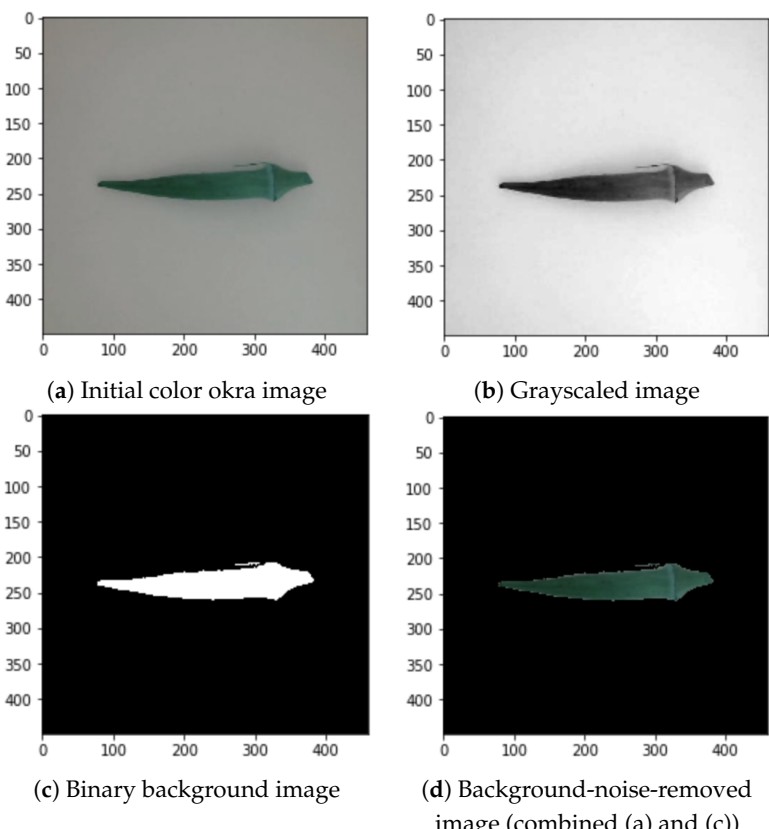

(**a**) Initial color okra image    (**b**) Grayscaled image

(**c**) Binary background image    (**d**) Background-noise-removed image (combined (a) and (c))

**Figure 5.** Background noise removal process.

After grayscaling the inputs and enhancing the pixels, we converted them into binary images [4]. The output binary image was subjected to area labeling to produce an identified region. Areas with pixel values above 0.46 out of 1 become equal to 1 (white), and the rest are equal to 0 (black). The value of 0.46 differentiates the foreground and background. Positive values were then substituted with their RGB counterparts. Figure 5c shows an example of the binary transformation and Figure 5d depicts the combination of the binary background and color okra image.

### 2.5. Effects of Color Space in Image Classification

Color spaces are widely considered in image classification [21–24] because they provide a variety of visual aspects, such as lightness, value, and saturation, which better depict certain features within images. We used color space transformation algorithms to generate other color spaces, such as HSL, YCrCb, and HSV, from the RGB dataset. We then compared the performances of these color spaces with that of the original RGB.

Hue, saturation, and lightness (HSL) are alternative color representations that can be obtained or deduced from RGB color space using conversion algorithms. This color space is represented by a cylindrical shape with a hue in the angular dimension, starting at 0° and ending at 360° in red (R). The starting and ending points were green (G) at 120 °and blue (B) at 240°. From the center moving horizontally to the edges is the saturation (S), also known as the intensity. This describes the degree of purity of hues. Throughout the vertical axis, the lightness (L) ranges from top to bottom, with white (value 0) at the top and black (value 1) at the bottom.

Hue, saturation, and value (HSV) are, similar to HSL, an alternative representation of RGB. As a color space, it is commonly used in machine vision and shares many similarities with HSL. It is represented by a half-cylindrical shape and is sometimes described as

the bottom half of the HSL. Its hue is in the angular dimension, starting from 0° at R and rotating by 360° to R. G and B are proportionally distributed in the middle at 120° and 240°, respectively. The saturation of HSV also describes the purity of the hue. The main differences between HSL and HSV are their lightness and value. This value can be considered as the quantity of light thrown onto an object. The complete presence of light turns an object white, whereas the total absence of light makes it dark or black. Compared to lightness, which measures white or black mixed with a pure hue (representing RGB), the value better represents the amount of light in an object. Thus, in some cases the color intensity of the objects was better.

YCbCr is a family of spaces widely used in image processing and classification. Y is the luma (brightness in an image) component to which the human eye is very sensitive, and Cb and Cr are the blue- and red-difference chroma components, respectively. Similar to the HSL color space, the values of Y, Cb, and Cr can be obtained through the mathematical conversion of the RGB values. This color space can provide a large spectrum of information regarding an image that cannot be accessed using the standard RGB color space. Moreover, as mentioned in a study on explicit image detection [25], YCbCr can achieve better object detection than RGB.

HLG is a combination of hue, lightness, and green. We consider this mix important because it considers many important visual features among the JA features. For example, green intensity and lightness are essential features because they differ between types A and B.

Using the Python library OpenCV, we created tuples containing each color space band and merged them with other color space bands to reproduce a new color space combination varying from three-dimensional to four-dimensional spaces. An example is the HRG image, which is made by merging H for HSL and R and G from RGB. The experimental results in the next section represent the best-performing color space mixes between RGB, HSL, and YCrCb.

## 3. Experimental Evaluation

In this section, we compare the accuracy metrics for different combinations of color spaces, image segmentation techniques, and datasets used in training and testing.

### 3.1. Summary of Experimental Results

Table 3 shows the experimental results for twenty-three cases from E1 to E23 in the column "Case E*x*", where the five color spaces RGB, YCbCr, HSL, HLG, and HSV were employed and two classifiers CNN and SVM were used. In the experiment, we utilized two datasets DS1 and DS2, shown in Table 2, for training and testing, as shown in the columns "Train DS*x*" and "Test DS*x*", respectively. We split the dataset into training and testing parts at a ratio of 80:20 when the same dataset was used for training and testing. We compared inputs without and inputs with segmentation based on *k*-means clustering for each combination indicated by the column "*k*6 Seg". We only dropped the backgrounds indicated in the column "BS" in the table for case E3 (Figure 5d). The table cells show the classification accuracy as a percentage and are colored with a heat map, with higher accuracy in deeper red and lower accuracy in deeper blue. White indicates moderate accuracy.

Cases E1–E5 in Table 3 show the accuracy of the classification of the combinations of either train DS1–test DS1 or train DS2–test DS2 with the CNN classifier. Meanwhile, cases E6 to E9 correspond to the results of the combinations of either train DS1–test DS2 or train DS2–test DS1. Similarly, for the SVM classifier, cases E10–E13 show the results of either train DS1–test DS1 or train DS2–test DS2, while cases E14–E17 correspond to results of the combinations of either train DS1–test DS2 or train DS2–test DS1. Finally, cases E18–E23 show the results when we use mixed datasets of DS1 and DS2 for training.

We used the ResNet-50+ model as a classification algorithm in the second experiment and tested it on the mixed datasets DS1 and DS2. ResNet-50 was initially trained only on RGB. Because RGB holds different information from other color spaces, only RGB images

would fit this model. Therefore, only three experiments were conducted using ResNet-50+, as listed in Table 4.

**Table 3.** Experimental results (accuracy for each experimental settings).

| Case Ex | Train DSx | Test DSx | BS | k6 Seg | Color Spaces | | | | | Model |
|---------|-----------|----------|-----|--------|------|-------|------|------|------|-------|
| | | | | | RGB | YCbCr | HSL | HLG | HSV | |
| E1 | | | | | 86.6% | 92.4% | 93.1% | 98.1% | 96.5% | |
| E2 | 1 | 1 | | ✓ | 54.4% | 97.8% | 100.0% | 95.0% | 92.6% | |
| E3 | | | ✓ | | 89.1% | 93.5% | 97.8% | 93.5% | - | |
| E4 | 2 | 2 | | | 98.0% | 93.4% | 97.4% | 98.0% | 95.0% | |
| E5 | | | | ✓ | 92.1% | 96.0% | 96.7% | 96.2% | 92.1% | CNN |
| E6 | 1 | 2 | | | 63.3% | 63.3% | 70.0% | 67.9% | 60.3% | |
| E7 | | | | ✓ | 67.0% | 67.0% | 70.0% | 80.0% | 62.8% | |
| E8 | 2 | 1 | | | 57.0% | 48.0% | 52.5% | 56.6% | 49.4% | |
| E9 | | | | ✓ | 48.0% | 48.0% | 49.0% | 48.9% | 49.8% | |
| E10 | 1 | 1 | | | 87.9% | 87.8% | 87.7% | 86.7% | 89.7% | |
| E11 | | | | ✓ | 89.4% | 87.1% | 91.1% | 90.5% | 90.1% | |
| E12 | 2 | 2 | | | 99.3% | 96.7% | 98.7% | 98.0% | 96.0% | |
| E13 | | | | ✓ | 97.4% | 96.0% | 94.7% | 91.4% | 91.5% | |
| E14 | 1 | 2 | | | 66.5% | 66.5% | 66.5% | 64.1% | 63.5% | SVM |
| E15 | | | | ✓ | 63.3% | 63.3% | 70.1% | 67.9% | 71.9% | |
| E16 | 2 | 1 | | | 48.0% | 48.0% | 48.0% | 51.0% | 48.0% | |
| E17 | | | | ✓ | 48.6% | 48.1% | 41.6% | 48.8% | 51.4% | |
| E18 | | Mix | | | 85.5% | 92.1% | 91.5% | 89.8% | 94.1% | |
| E19 | Mix | 1 | | | 95.4% | 93.7% | 97.5% | 97.6% | 97.6% | CNN |
| E20 | | 2 | | | 78.4% | 86.1% | 84.6% | 83.6% | 80.5% | |
| E21 | | Mix | | ✓ | 88.7% | 90.5% | 90.0% | 89.1% | 90.1% | |
| E22 | Mix | 1 | | ✓ | 96.4% | 96.6% | 98.2% | 96.7% | 96.6% | CNN |
| E23 | | 2 | | ✓ | 78.2% | 72.4% | 82.9% | 86.4% | 79.7% | |
| Avg. for all cases | | | | | 77.3% | 79.3% | 81.3% | 81.6% | 79.1% | |
| Avg. except for different DSs | | | | | 87.8% | 91.5% | 93.5% | 92.7% | 91.6% | |

**Table 4.** Evaluation of RGB color space with the ResNet-50+ model (mixed dataset, DS1, and DS2).

| Case Ex | Train DSx | Test DSx | Color Spaces | | | | | Model |
|---------|-----------|----------|------|-------|------|------|------|-------|
| | | | RGB | YCbCr | HSL | HLG | HSV | |
| - | 1 | 1 | 90.9% | - | - | - | - | |
| - | 2 | 2 | 95.4% | - | - | - | - | ResNet-50+ |
| - | Mix | Mix | 92.9% | - | - | - | - | |

### 3.2. Comparison

This section compares the results shown in the above table from the five viewpoints: color spaces, region-based segmentation, background-segmented images, datasets, and classification algorithms.

### 3.2.1. Color Spaces

After comparing five color spaces over 110 experiments, we observed the significance of the color space in image classification. In most comparisons between the CNN and SVM models, the RGB color space had lower accuracy scores than the other color spaces. For example, if we analyze cases E1 to E3 in Table 3, RGB scores approximately 10% less than the best performant color space for cases E1 and E3. In the case of segmented images, E2 achieved a score that was over 40% lower. However, RGB performed as well as the other

color spaces when DS2 was used as the training set in cases E4 and E5 and, in some cases, slightly better.

The most effective color spaces for classifying okra were HSL and HLG. In cases E1–E7, E11, E14, E16, E19, E22, and E23, these two color spaces achieved the highest scores and approximately a 10% improvement in some cases compared with the standard RGB color space. Moreover, even in cases where HSL and HLG did not perform the best, the difference in accuracy between these color spaces and the best-performing color space was not significant. Thus, by comparing the color spaces of multiple algorithms and training datasets, we can confirm that color space transformation is highly beneficial for good image classification. HSL and the combination of HLG emphasize lightness and color intensity variations better than the other color spaces. Essential features, such as color and the presence of dots, are better depicted by these colors, which could potentially explain why they generally perform better.

### 3.2.2. Region-Based Segmentation

As with the color spaces, we analyzed the effectiveness of image segmentation in solving image classification problems. We confirmed that the effects of image segmentation vary depending on the training dataset based on the results presented in Table 3. When DS1 was used for training, the use of segmentation led to a better accuracy in classification compared to the results obtained without segmentation. Conversely, the classification without segmentation achieved a better accuracy when DS2 was used for training. The raw data for DS1 and DS2 were obtained from different products but from the same farm on different dates. From a practical perspective, different datasets obtained under different conditions must be used. Therefore, we need to use segmentation carefully as a preprocessing step according to the characteristics of the datasets.

### 3.2.3. Background-Segmented Images

In many studies, the segmentation of the image background has often been correlated with a higher accuracy. This technique helps reduce the amount of background noise. Therefore, the less noisy the image, the better the classification. However, segmenting backgrounds using the different techniques in this study did not prove effective. It always performs lower than $k$-means clustering. Therefore, we drop the background segmentation at the early stages and only maintain $k$-means clustering.

### 3.2.4. Datasets

We achieve a good classification accuracy when we use DS1(DS2) as a training set and test it on unseen DS1(DS2) data. However, the train DS1–test DS2 or train DS2–test DS1 often led to low scores, as shown in cases E6-E9 and E14-E17. To address this issue, we used the mixed dataset. As explained above, the mixed dataset contains both the DS1 and DS2 datasets. Thus, the mixed dataset addresses the subtle differences between the two categories. The results of the experiments show that using the mixed dataset for training resulted in high scores on tests DS1, DS2, and mixed. Moreover, the mixed dataset did not require as many images as the training set. Even when the training set was smaller than the testing set, it scored over 90% for almost all categories. Furthermore, it still performed better than the train DS1–test DS2 and train DS2–test DS1 even when the training size was as small as 20% of the size of the testing dataset.

### 3.2.5. Classification Algorithms

In general, although by a small margin, the CNN classifier outperforms the SVM classifier. One of the main reasons for this performance difference is that the CNN classifier can consider the color space features and segmentation properties, which the HOG-SVM cannot. We utilized HOG-SVM to accelerate the process and focus only on vital details. This technique ignores region-based segmentation and only works with what is judged as necessary throughout the HOG transformation. However, it is crucial to note that the

SVM classifier performed as well or slightly better than the CNN when DS2 was used as the training set.

By combining the results of 123 experiments, we confirmed that color spaces are important in image feature detection and classification. In particular, hue, lightness, and saturation performed better than other color spaces. Nonetheless, a color space combination, such as HLG, can sometimes perform better than HSL. Thus, color space transformation is necessary when dealing with data with a high emphasis on color intensity and different lighting conditions, and combining color features can be more efficient. Furthermore, not only do color spaces matter, but they also affect the classification better when combined with region-based segmentation because this technique helps to split the foreground, background, and necessary and unnecessary information. The results indicate that the CNN is a better-fitting algorithm than the SVM for this case study.

## 4. Discussion

Throughout this study on image preprocessing and classification, we demonstrated the effectiveness of using machine learning and image processing techniques to accurately detect and classify vegetal species.

In the post-harvest okra classification, we observed some factors that contributed to better classification. First, the color space transformation correlates with more feature detections. Color spaces are not always considered in the image classification literature. We agree that in some cases, such as the HOG-SVM classification, the transformation of the color space does not have any noticeable difference. By contrast, this only lengthened the preprocessing time. However, under similar conditions as the okra study, in which lightness and color intensity are considered, it is valuable to execute color space transformation, particularly the transformation to HSL, HSV, or, as in this study, a meaningful mix between color spaces. Compared to the RGB color space, other color spaces led to higher accuracy scores. Another important factor in this study was the background segmentation of images or segmentation through *k*-means clustering. Combined with specific color spaces, it segments the foreground from the background and extrapolates lightness and color-related features. Therefore, combining this with a CNN classifier is valuable.

The CNN models performed better than the SVM models. We use a histogram of oriented gradients (HOG) within our SVM models during preprocessing. This method obliterates all features obtained through color space transformation. This result explains why grayscaling is required to obtain HOG features. Moreover, the grayscale images showed no noticeable differences between the color spaces. Thus, we can assume that color space transformations are unnecessary when using HOG-SVM models. As mentioned earlier, this process is time-consuming. However, a crucial finding is that the SVM classifier outperformed the CNN classifier when the dataset and classes were small. As shown in cases E12 and E13 of Table 3, the SVM model almost perfectly classified the DS2 dataset, regardless of the color space. This result can be explained by CNNs needing to be trained with diverse and large datasets to perform well. However, the HOG-SVM classifier reduces the complexity of the problem while maintaining as much variation as possible. In this sense, HOG-SVM classifiers seem less dependent on dataset size than CNN classifiers.

One of the reasons why the okra study tests the DS1 dataset on the DS2 dataset and vice versa and the combination of both datasets is to evaluate the performance of the models on different unseen data. However, DS1 and DS2 are considered to bear only minor differences in appearance. Nonetheless, the accuracy scores were quite low when DS1 was used as training for DS2 testing and DS2 was used for DS1. The divergence in their crucial features may explain this discrepancy. To solve this problem, we propose the following solutions:

1. The first solution is to prepare training datasets with enough divergence. It can be accomplished by carefully combining different datasets. In our study, for example, we prepared a mixed dataset containing 758 images when DS2 was the target for testing. Even with small numbers for training, say 80% of it, we achieved a good performance,

as shown in Table 3. The differences between DS1 and DS2 are seemingly small but very critical. This demonstrates the importance of matching certain conditions and diversifying the training data as much as possible. Moreover, it exhibits the possible superiority of diversity oversize. Although the training dataset needs to be large enough, it is equally essential for the dataset to be diverse enough. This fact is theoretically straightforward, but it is not easy to achieve diversified datasets practically.

2.  The second solution involves implementing fine-tuning methods. As shown in Table 4, the ResNet-50+ model achieves good results in the three datasets (mixed, DS1, and DS2), despite not having been initially trained to classify okras. The initial pretrained model was used to classify over 1000 categories of images from a million images. However, when used with its initial weights and layers, it performs poorly in the binary classification of post-harvest okras. After adding a few convolutional layers, as described in the okra classification method, the model achieved high accuracy. ResNet and the other pretrained models contained a large amount of information and were trained to detect features in over 1000 non-related categories. Therefore, using such models can help reduce performance and time costs, particularly when not using "high-spec" devices because deep learning is a resource-intensive task.

By utilizing different and diverse methods, we achieved fairly good results for post-harvest image classification; however, we still have plenty of room for improving classifiers and preprocessing techniques. Therefore, we propose the following for future studies.

### 4.1. Better Image Quality and Details Enhancement

The okra datasets were divided into two groups: DS1 and DS2. The photographs of the two datasets were organized under slightly different conditions, with different okra dimensions, lighting conditions, and positions. Moreover, noise did not affect the datasets in the same way, and okra were not centered on many images, requiring us to execute time-consuming preprocesses to prepare the image for training efficiently. Finally, the image quality did not allow for the detection of black dots on the okra surface. The presence and number of black dots are some of the JA's classification features. Because of these factors, the models built with DS1 could not be classified with high-accuracy DS2 and vice versa. Such a situation is common from a practical perspective. Therefore, we must use datasets obtained under different conditions. In future work, we will conduct more experimental evaluations with more diverse datasets and propose an efficient design of mixed datasets for training.

### 4.2. Testing in Real World

Using multiple preprocessing techniques and image segmentation through *k*-means clustering, we achieved a perfect accuracy score for classifying the DS1 okra dataset. Furthermore, the use of HSL and HLG color spaces led to higher classification accuracies than RGB. In future work, we will consider implementing these models, techniques, and other product classifications, as well as deploying and testing our models in real-world scenarios.

### 4.3. Building a Better-Performing Model

The CNN models designed in this study considered myriad factors. We avoided overfitting and underfitting while training the models, shuffled the dataset at each iteration to avoid bias, proceeded with data augmentation, and compared over 50 CNN architectures. We compared this algorithm with the SVM algorithm and other popular CNN models. However, there are multiple ways to improve the model. One method that we will consider in future work is to implement and improve pretrained models, such as VGG19.

## 5. Conclusions

In this study, we designed and implemented machine learning solutions for the post-harvest classification of okra. In our approach, we carefully created training datasets from raw images. Raw data were obtained under different dates, times, and weather conditions.

Therefore, the image data included biased shades, noise, and ambient light variations. In the experiment, we used two different datasets, DS1 and DS2, and their combination (the mixed dataset). Furthermore, we preprocessed the dataset using numerous image processing techniques to depict the variations and features within each okra class. The preprocessing included further cropping of the okra background, segmenting images with *k*-means clustering, changing the color space, and resizing and reshaping. We then applied the three algorithms as classifiers to compare the best-performing algorithms, considering their accuracy scores. The classifiers included a CNN that we developed for this purpose, a HOG-SVM classifier, and a fine-tuned neural network. We found that, by considering color spaces, such as HSL and HSV, and image segmentation, we achieved accuracies of 90.0% for testing the mixed dataset, 98.2% for DS1, and 86.4% for DS2 with our CNN when using the mixed dataset as a training set. The benefit of using the mixed dataset was that it required less training data than the other datasets, provided that it had sufficient diversity. Furthermore, it performed well in classifying all three sets (DS1, DS2, and mixed dataset).

When the image data in DS1 were utilized for training, almost 100% accuracy was achieved in classifying unseen data in the same DS1 (with HSL color space image and segmentation). However, it scored only 80% when classifying data in DS2 using the HLG color space. Finally, when DS2 was utilized for training it achieved 99.3% (with RGB) accuracy on itself and only 57% (with RGB) for testing DS1 at best. Thus, we emphasize the importance of diversifying the datasets and changing the color spaces. We also mentioned the benefit of implementing a fine-tuned deep learning algorithm (ResNet-50+) to classify okra because it achieves 90% accuracy on all datasets and requires only a few layers to be trained. In conclusion, while comparing the accuracy metrics, fine-tuning was found to be the best-performing method for classification while considering the time cost. However, the diverse mixed dataset obtained using our CNN model provided the most accurate predictions.

Machine learning (ML) is a promising solution for smart agriculture. However, there are many issues to be solved in real farm applications owing to their uncertainty. This paper presents essential points for AI practitioners in agriculture to design and develop their post-harvest classification effectively. In future work, we will develop an automatic parameter-tuning technique with lower computational costs to enhance its applicability on real farms.

**Author Contributions:** Conceptualization, M.N. and Y.N.; methodology, P.M.D. and J.T.; software, P.M.D. and J.T.; validation, N.O. and M.N.; formal analysis, P.M.D. and M.N. All authors have read and agreed to the published version of the manuscript.

**Funding:** This research received no external funding.

**Data Availability Statement:** The data used in this study are available from the corresponding author on reasonable request.

**Conflicts of Interest:** The authors declare no conflict of interest.

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
