# Peer review of "Design of Machine Learning Solutions to Post-Harvest Classification of Vegetal Species"

_agriengineering, doi:10.3390/agriengineering5020063_

Round 1
Reviewer 1 Report
This paper proposes a novel approach to classify post-harvest vegetal species using machine learning techniques. The study focuses on okra and classifies it into two quality types using color images of the vegetable. Besides, the proposed approach can be applied to other species as well.
The authors have used convolutional neural networks (CNNs) and support vector machine (SVM) classifiers to classify the images. The color spaces used in the study are HLG and HSL, which are based on combinations of hue, lightness, and green factor for HLG, and hue, saturation, and lightness for HSL. The study has shown that color spaces are crucial to these classification problems.
The authors have also used effective preprocessing techniques due to the complexity of the data and noise load. Moreover, to reduce the possibility of overfitting, the authors have created a training dataset by combining different datasets obtained under different conditions.
As presented in the Introduction part, it can be concluded that deep learning is playing an important role in automatic detection and classification. A short introduction on the application of deep learning techniques in different areas should be presented, and some relevant examples are: doi:10.3390/app12094356; doi:10.1007/s00170-022-10335-8.
The experimental evaluation confirms that the proposed approach is effective in classifying okra into two quality types. The results show that the color spaces and preprocessing techniques used in the study are adequate for vegetal species classification.
Overall, the paper is well-written and presents a novel approach to classify post-harvest vegetal species. The authors have used appropriate machine learning techniques and preprocessing techniques to achieve the desired results. The experimental evaluation provides evidence of the effectiveness of the proposed approach. The study's findings can be valuable for the food industry, as it can help classify different species based on quality, which can enhance the efficiency and productivity of the industry.
Overall, the quality of English language in the paper is satisfactory. The paper is well-written for a scientific research paper, and the authors have used technical terms and jargon effectively.
Author Response
Dear Reviewer1
Thank you very much for your careful reviewing our paper and valuable comments.
According to the reviewer’s comments, we added one reference from the recommended list to focus on agriculture applications in this paper.

Reviewer 2 Report
In the abstract, please explicitly state your study's main implications. Who wins what and why upon your findings?
Please strongly avoid lumped refs, as they are almost useless for the audience.
In ... can deal with all these factors ... to what the word these refers? Please avoid uncertainty in making references.
At the end of the introduction, please formally state what the research gap your study aims to bridge is, and what are the research question, purpose, and research method (lab experiment) you employed to overcome the gap.
Please avoid third level indentation 2.2.1, two levels are enough. Reorganize the text if necessary to comply with two levels.
Small subsections in chapter 3 should be merged in a single one.
The experiment is fine. Nonetheless, if possible, some type of sensitive analysis could be done. What would happen if some initial parameters slightly varies, by for example 5%? If it is viable, please improve the text.
Conclusion should focus mainly on implications. Whats next? Please answer briefly who wins what and why upon your findings? Are there some environmental or economic gain or benefit? To whom?
Best regards
Author Response
To Reviewer 2:
Thank you very much for your valuable comments. We improved our paper according to the comments. So, could you please check our revised article?
- In the abstract, please explicitly state your study's main implications. Who wins what and why upon your findings?
We rewrote the abstract to state the main contribution of our research.
- Please strongly avoid lumped refs, as they are almost useless for the audience.
Yes, we agree. We improved the references.
- In ... can deal with all these factors ... to what the word these refers? Please avoid uncertainty in making references.
We improved the sentences to clarify the refers.
- At the end of the introduction, please formally state what the research gap your study aims to bridge is, and what are the research question, purpose, and research method (lab experiment) you employed to overcome the gap.
Yes, we improved the introduction to state the main contribution.
- Please avoid third level indentation 2.2.1, two levels are enough. Reorganize the text if necessary to comply with two levels.
According to the comment, we reorganize the text.
- Small subsections in chapter 3 should be merged in a single one.
Yes, we merged small subsections in Section 3.
- The experiment is fine. Nonetheless, if possible, some type of sensitive analysis could be done. What would happen if some initial parameters slightly varies, by for example 5%? If it is viable, please improve the text.
We experimented with varying parameters and chose a set of reasonable ones. However, we omit to show the sensitivity analysis. In our future works, we will show extended results with sensitivity analysis.
- Conclusion should focus mainly on implications. Whats next? Please answer briefly who wins what and why upon your findings? Are there some environmental or economic gain or benefit? To whom?
We improved the conclusion section to state our contribution and future tasks.

Round 2
Reviewer 2 Report
OK to publication